# Factors That Support Technology Integration Practices in the Math Education of Children with Intellectual Disabilities

**DOI:** 10.3390/children10060918

**Published:** 2023-05-23

**Authors:** Carmen Viorica David, Cristina Anamaria Costescu, Adrian Marian Roșan

**Affiliations:** 1Academy of Romanian Scientists, Splaiul Independenţei, 050094 Bucureşti, Romania; christina.costescu@gmail.com; 2Special Education Department, Faculty of Psychology and Educational Sciences, Babes-Bolyai University, 400084 Cluj-Napoca, Romania; adrian.rosan@ubbcluj.ro

**Keywords:** digital technologies, technology-related knowledge, teachers’ beliefs, intellectual disabilities

## Abstract

The current paper explores the relationship between technological knowledge, beliefs about technology use in education, beliefs about the limits of technology use with respect to children with disabilities, self-efficacy beliefs, and their effects on technology use or teacher practices in the special education setting regarding mathematics education. Research was conducted via an online questionnaire. A total of 83 teachers working in the field of special education answered the questionnaire. The answers were analyzed via quantitative and qualitative analyses. In addition, correlational analyses were conducted. A prediction model that included all the aforementioned variables was tested. Our regression analysis evidenced the strong predictive value of technology-related knowledge with respect to technology integration practices but not toward the other models that included beliefs. Through mediation analysis, we found that technology-related knowledge mediates the relationship between self-efficacy and technology integration practices. The qualitative findings provided information on the interplay between knowledge, practice, and beliefs that are contextualized; on the specificity of technology-related knowledge. Implications are discussed with reference to factors important for the professional development of teachers with regard to technology integration.

## 1. Introduction

In the past two years, with the prevalence of online schooling, integrating technology into teaching has become important and gained more consistency. It has become critical for a teacher to be able to use technology as a fully instructional tool for learning and not just as a supplement for instruction. Moreover, given the advantages of many technologies, they constitute great and accessible means through which special education teachers can individualize [1,2] and even personalize instruction. However, while research on the factors that support the integration of technology into teaching and learning in the case of general education teachers is abundant, research focusing on special education is not so prolific. Even more scarce is research on the integration of technology in the math education of children with special educational needs.

Using a qualitative methodology, Baglama and colleagues [3] addressed 15 special education teachers’ views on using technology when teaching mathematics. The results showed that the teachers felt competent using technology in the classroom. Most respondents indicated that they employ mathematical games from the internet, a similar number reported using general Office software, only five claimed to use specific software for education, and two reported that they did not use technology for teaching. One interesting result was a view that was reported by four teachers from the participant group on the problems encountered when using technology in teaching mathematics. They felt that “excessive use of technology might lead to addiction” (p. 127). The study did not address the relationship between perceived competency in using technology for teaching math and the use of technology in the classroom.

Cagiltay and colleagues [2] conducted a qualitative study on the perceptions of teachers regarding educational technologies. One interesting result was that while they reported a positive interest in using technology in their teaching, they were not so open to new technological developments and updating knowledge (unless mandatory). The authors related this finding to the lack of technology and materials available for special education teachers as well as financial barriers.

Knowledge on evidence-based practices regarding the use of technology in math education or math intervention for children with special needs is accumulating. It was indicated Spooner and colleagues [4] that the efficacy of using manipulatives and technology-aided instruction. Another study indicated a moderate effect of interventions based on digital strategies for children with specific learning disorders [5]. In addition, they underlined the high variability across studies and the need to further examine factors that relate to intervention efficacy. This scarce research and the variability in the research results may contribute to the limited responses of teachers with respect to updating knowledge. 

## 2. Literature Review

### 2.1. Relationship between Technology-Related Knowledge and Technology Use Practices in the Classroom

Teachers’ knowledge is difficult to define [6] and is conceptualized in several ways. Throughout this study, we will employ the term technology-related knowledge [7] which broadly refers to knowledge on the adequate use of technology for the purpose of teaching and learning. In a more specific sense, we refer to the knowledge necessary to be able to integrate technology, as indicated [8]: a. knowledge on the content and pedagogical affordances of technology; b. knowledge on adequate uses of technology with respect to the learners; and c. knowledge on existing technological tools used for various teaching and learning purposes. Technology-related knowledge has been conceptualized in several ways. One of the most frequently employed models in the literature is TPACK [9]. In this model, a teacher’s critical knowledge is segregated into three different areas: content knowledge (knowledge of a subject), pedagogical content (knowledge on teaching and learning), and technological content (knowledge on technologies). This framework offers a model of knowledge’s integration into more elaborate knowledge systems and highlights two levels of integration: in between each knowledge area and in all three areas. A factor analysis conducted [10] on a survey built based on the TPACK framework. Their results indicated that items concerning technological content, technological pedagogy, and technological pedagogical content loaded on the same factor, while the items regarding technological knowledge loaded on different factors. Another factor analysis was conducted [6] on a TPACK instrument administered to preservice teachers in Estonia. Their results indicated a three-factor structure for which all items that dealt with technology were loaded on the same factor.

It was underlined the importance of a teacher’s knowledge to making educational decisions, including with respect to the integration of technology into the classroom [11]. In a review of studies published between 1995 and 2006 [12], it was pointed out that “the lack of specific technology knowledge and skills, technology-supported pedagogical knowledge and skills, and technology-related classroom management knowledge and skills has been identified as a major barrier to technology integration” (p. 227). In addition, it was analyzed the impact of professional knowledge on technology integration [13]. A direct effect of knowledge on technology and its integration was evidenced with respect to technology integration, while pedagogical knowledge only had a complete effect on technology integration through the mediation of knowledge of technology and its integration. 

There are some challenges encountered when measuring technology-related knowledge. With regard to measuring knowledge concerning the use of technology in teaching, self-reporting tools, performance analysis of teaching-related activities, and scarce knowledge tests are employed [7]. One major critique concerning the measuring instruments used is that beyond the social desirability of self-reporting, many questionnaires contain items that, rather than measuring knowledge, measure perceived self-efficacy in using technology for teaching and learning [7,13,14,15]. 

### 2.2. Relationship between Teachers’ Beliefs and Technology Use in the Classroom

Affective factors also play a major role in decision-making processes, including those related to integrating technology into teaching and learning. Among these, beliefs are mostly researched in terms of their content and implications for technological integration. A definition provided Hsu and colleagues [16] which is based on the work of Pajares and colleagues [17], states that beliefs are subjective constructs that filter as lenses, help teachers interpret experiences, and are instrumental to teacher’s actions, decisions, and approaches in the classroom. However, the content of beliefs is broad [13,17,18], even within the realm of educational beliefs [17]. Beliefs about technology integration must also be considered, for which the following subcategories have been specified: beliefs that correspond to a pedagogical orientation, self-efficacy beliefs, and valuative beliefs about technology [19]. Beliefs, which constitute a component of a belief system, have different bearings in relation to other beliefs or other cognitive or affective structures [17]. Therefore, it is expected that some beliefs play a greater role in decision-making processes than others. 

Many studies have provided evidence for the importance of teachers’ beliefs in integrating technology [11,18,20,21,22]. On the other hand, some studies found that teachers’ beliefs did not define technology use in the classroom [23]. McCulloch and colleagues [24] analyzed the content of beliefs that played a role in secondary math teachers’ selection and integration of a particular technology into a math class. Their beliefs about technologies and students’ learning played an important role at different stages of the decision-making process. It was evidenced that teachers’ pedagogical beliefs played an important role in technology’s integration and use in the classroom through their impact on beliefs about the value of technology use in the classroom [13,25]. Positive beliefs about the value of technology in the classroom are associated with an increased degree of adopting technology integration for educational purposes [26,27]. Moreover, Hsu and colleagues [16] evidenced an association between a teacher’s pedagogical beliefs, self-efficacy beliefs, valuative beliefs, and the integration of technology into the classroom.

Digital technology acceptance is a relevant factor with respect to the integration of technology into the classroom. There are several models concerning the integration of information and communications technology (ICT), of which the most popular is the technology acceptance model [28,29]. According to this model, technology acceptance is governed by two variables: the perceived ease of using technologies and the perceived usefulness. There are many ways of investigating this subjective perception. For instance, [30] delineated a typology based on the attitudes towards new media, namely, techno-pessimist, techno-optimist, techno-realist, and techno-ignorant. In a recent paper, a group of researchers investigated teachers’ opinions and experiences regarding the implementation of new technologies in the classroom [31]. Their study involved 873 teachers from 8 different countries and yielded important findings regarding techo-optimism. Even though there are different approaches among different countries, when analyzing the data from the survey, teachers generally tended to be techno-optimists. On the other hand, it was mentioned the fact that the use of information and communication technologies can cause teachers stress (technostress), which can influence their well-being, especially in terms of their emotional state [32]. The authors’ sample consisted of 1210 teachers, for which the majority came from Italy, and their findings suggest that teachers may experience negative emotions while practicing distance learning and that the risk of technostress (with three related domains: anxiety, fatigue, and ineffectiveness) can predict burnout among teachers. However, personal differences must also be considered when investigating the use of technology among teachers. Experience, for example, can be a great predictor of technology acceptance among teachers, as shown [33]. Their results show that teachers with 21 years of experience or more seem to be more unlikely to adopt technological tools within their classrooms compared to teachers that are less experienced.

### 2.3. Relationship between Self-Efficacy Beliefs and Technology Use

Self-efficacy beliefs about the integration of technology into teaching and learning concern teachers’ perceptions of their capacities and the confidence they have regarding their abilities to learn about and use technology for educational purposes [16]. These types of beliefs are strong predictors of technology use and integration [18,34]. Results have indicated that self-efficacy is related to a more extensive use of technology or to the integration of technology in teaching and learning [35]. On the other hand, another study did not obtain evidence on a direct effect of self-efficacy beliefs regarding the use of technology and technology integration in a sample of teachers [13].

In the special education context, the topics of self-efficacy beliefs in the use of technology and the relationship between self-efficacy beliefs and the integration of technology into the classroom have been less researched [27,34]. A team of researchers analyzed the relationship between confidence with using technology, teaching experience, teachers’ beliefs, and TPACK knowledge [36]. Teachers that were confident with technology tended to integrate technology into their teaching and learning practices to a greater degree.

## 3. Purpose

The purpose of the current study is to investigate special education teachers’ technology-related knowledge; self-efficacy beliefs; beliefs about the use of technology for education, specifically in the case of children with intellectual disabilities; and practices regarding the integration of technology into the classroom in math education. In addition, we aim to investigate the interplay between beliefs held by special education teachers about technology use and their self-efficacy beliefs regarding the use of technology in order to integrate more technology into current practice. 

The hypotheses and research questions examined in this study are as follows:

**H1.** 
*Teachers’ technology-related knowledge predicts their technology use in the classroom.*


**H2.** 
*Teachers’ beliefs about the value of technology predict their technology use in the classroom.*


**H3.** 
*Teachers’ self-efficacy beliefs predict their technology use in the classroom.*


**H4.** 
*Teachers’ beliefs about the limits of technology with respect to the education of children with intellectual disabilities predict a limited use of technologies in the classroom.*


Given the complex system of beliefs and the possible different weights of such beliefs with respect to the decision-making process regarding the integration of technology in math education for children with special needs, we added a research question to these predictions:

RQ. What are the relationships between technology-related knowledge, self-efficacy beliefs, valuative beliefs, and beliefs about technology’s limits and technology use in the classroom?

## 4. Materials and Methods

### 4.1. Participants

In our research, we used a convenience sample consisting of teachers working with children with intellectual disabilities. We recruited and invited the respondents via informative online sessions or offline description of the research that was presented to teachers from special schools during professional development meetings via internal social media groups or other institutional informative channels. Each of the respondents agreed on the informed consent before completing the on-line questionnaire. No incentives were given for participation. Eighty-three teachers responded to our questionnaire, of which seventy-eight were females (94%). We included specialists from several administrative counties who specialized in special education and working with students with intellectual disabilities. A total of 51.8% (N = 43) had attained the highest qualification level in teaching, 24.1% (N = 20) had attained the second highest degree of qualification, and a similar percentage (24.1%) had entry-level qualifications. A total of 48.2% of the respondents were between 41 and 50 years old, 32.5% were between 31 and 40 years old, 13.3% were between 50 and 60 years old, and 6% were between 20 and 30 years old. A percentage of 66.3 worked with students that present mild and moderate intellectual disabilities, while 51.8% worked with students with profound, severe degrees of intellectual disability and associated disabilities [37]. 

Respondents taught at self-contained middle school grade (32.5%), elementary school self-contained grade (34.9%), multiple grades at the middle school level (7.2%), and multiple primary school grades (2.4%), while 22.9% taught students with special needs from elementary through eighth grade in special settings. A total of 14.5% of the respondents were resource teachers (teachers that offer support to children in inclusive schools). Most of the respondents had more than 20 years of teaching experience (36.1%), 10.8% had between 16 to 20 years of teaching experience, 16.9% had between 11 and 15 years of experience, 20.5% had between 6 to 10 years of experience, and 14.5% had at most 5 years of experience. A total of 71.1% of the participants had taken between 1 to 3 courses on applications of digital technologies in education and 19.3% did not, while 9.6% reported having participated in more than 3 courses. Specific training courses on applications of digital technologies in math class were completed by 32.5% of the respondents, while 67.5% did not take any courses on this topic.

### 4.2. Procedure

This study was conducted in accordance with the ethical standards required by the Ethical Committee of Babeș-Bolyai University. The questionnaire was administered via email through Google forms between May and August 2022.

### 4.3. Questionnaire Development

The questionnaire aimed to identify the types of digital strategies used, the frequency with which they are used, and teachers’ knowledge and beliefs about the use of different technologies. The section on self-efficacy beliefs was developed based on the following research: *Survey of Preservice Teachers’ Knowledge of Teaching and Technology* [38] (main source); [39,40,41]. Many instruments assessing technology self-efficacy are technical [42] or general. Beliefs about the value of technology and its limits in education were addressed based on the research conducted [36,40]. An element of novelty in this study is the customization of the questionnaire’s content with respect to the content and competencies of the special education curriculum as well as the inclusion of content related to knowledge about digital technologies for use in mathematics education of self-contained children with intellectual disabilities.

Many instruments used to assess technological pedagogical content knowledge have been proven to address self-efficacy instead [7,13,14,15]. For this reason, we did not use a standardized questionnaire. In addition, another recommendation when assessing practices of technology integration is to gather data on the frequency of use as well; this practice was followed in our research. 

This instrument has been developed to address specific technology integration knowledge regarding the mathematics education of children with intellectual disabilities, as many instruments require such knowledge to be self-rated. In addition, we developed a scale to address technology integration practices that incorporates more information than the frequency of use. Furthermore, while attempting to refine the types of beliefs that play a role in technology integration, we formulated valuative beliefs and separated them from beliefs about the limits of technology. Self-efficacy beliefs were formulated by using *Survey of Preservice Teachers’ Knowledge of Teaching and Technology* [38] (main source) as a starting point. The content was contextualized to teaching math with technology. 

Regarding the technology-related knowledge scale, we used open-ended questions and a matching item to address the technology-related knowledge of teachers. Three questions addressed knowledge of digital technologies used for the math education of children with intellectual disabilities (e.g., enumerate some advantages of using interactive PowerPoint for the math education of children with intellectual disabilities), three questions addressed pedagogical uses of technology in math education (e.g., give an example of technology that can support the formation of mathematical representations), and two questions addressed knowledge of different types of technologies for learning (e.g., match the following resources with the apps/software you can use to create them: conceptual maps/educational movies/electronic presentation/interactive activities for practice/Prezi/Storyjumper/Mindmeister/Twinkl.).

Regarding the technology integration practices, we used questions with response alternatives. All questions address what was used or created with digital technologies. “When returning from online school to on-site, I have used the following technologies: Educational software/Office software/Mobile apps/Tablet apps/Online activities in Wordwall, LearningApps/Interactive activities on the smartboard/e-learning platforms/Augmented reality/other”.

Valuative beliefs about using technology in the education of children with intellectual disabilities were selected; the content included specific uses of technology for instruction and assessment, type of content, stages of a lesson, etc. 

Regarding beliefs about the limits of using technology in the education of children with intellectual disabilities, the content was derived from research that addressed this topic through interviewing special education teachers. 

Self-efficacy beliefs were formulated based on the work of [38] in order to address different types of integrated knowledge when using technology. The first item addresses self-efficacy in using technological pedagogical content knowledge. Item 2 addresses self-efficacy in using technological pedagogical knowledge in teaching. Items 3 and 4 address self-efficacy in using technology for teaching math in the case of children with intellectual disabilities. The last item addresses self-efficacy in improving knowledge and skills related to technological integration in education.

#### Stages in Developing the Questionnaire

Discussion with an expert group of higher education teachers (N = 3) was conducted to circumscribe the themes:Analysis of curriculum content and curricular competencies.Consultation of other questionnaires with the objective of assessing knowledge, practices, and beliefs.Consultation of specialist works on digital technologies used in education in order to circumscribe categories.Compilation of a bank of items for each theme.Discussion regarding the items for each theme.Verification of the grouping of items around the major themes (via a rating scale procedure); the items developed around technology-related knowledge and technology integration practices were further rated by 3 experts (an experienced teacher who was also a PhD student; a human resources expert with a PhD; and a highly qualified special education teacher).Acquisition of the final form for application.

The development of the questionnaire was based on existing works in the field. However, we needed a more contextualized investigation instrument that would allow for the measurement of our variables. Previous indicated work was consulted and constituted the basis for the development of the items. We used contextual information, and for this reason, we consulted special education math curricula. In addition, we considered a wide range of digital technologies in order to obtain a good sample. Each item was further analyzed with respect to its content and its categorization within a particular construct. The items that were not clear were reformulated, while those that were redundant were eliminated and those that did not fit the construct were either reformulated or eliminated. Once we decided on the items that remained, we further analyzed content validity via a multi-step procedure. We provided the experts with a clear definition of the construct. We used an analysis checklist in which we introduced the items concerning technology-related knowledge and technology integration practices. Experts were asked to indicate the category in which each item belonged and whether the formulation was clear. A good level of agreement was obtained for most of the items but not for Practice 7. We discussed this result with the experts; afterwards, the expert reconsidered the category. One item on the Practices subscale was eliminated, as it was not clear in terms of its formulation and did not generate agreement. The remaining items were included in the final form of the questionnaire.

Our questionnaire comprised a total of 55 items, which were organized into the following sections: informed consent and demographic information, technology-related knowledge, technology use practices, and beliefs (beliefs about the value of technology in education, technology limits in the case of students with disabilities, and self-efficacy beliefs about technology) (Table 1); an additional section that was not included in this research inquired about perceived barriers and facilitators of the integration of technology into the classroom. This part was purposely excluded but will be included in another article focusing on barriers and facilitators in technology integration. The questionnaire included open-answer questions and multiple variants. All answers to the open-ended questions were rated based on clear criteria (see Appendix A).

### 4.4. Measures

We measured the internal consistency of the questions that addressed the three different types of beliefs as well as technology-related knowledge and technology integration practices. For this purpose, we ran a reliability analysis using alpha Cronbach method in SPSS 20 (IBM SPSS Statistics for Windows, Version 20.0., Armonk, NY, USA: IBM Corp.). Table 2 represents the reliability statistics for each of the three measures.

These values indicate the good reliability of the four measures. The reliability analysis yielded a lower value for technology integration practices; however, this value was still acceptable. This lower value might be the result of using various types of questions, such as open-ended questions or those with alternative answers, e.g., asking for a percentage estimation or for a specific type of digital technology. One explanation for the lower value of the reliability coefficient is that the measure includes only three items. Another explanation is related to the content of the items, as each item samples beliefs on the limits of technology. For this reason, the correlation between the items may not be as high.

### 4.5. Analysis

In order to test our hypotheses and explore our research questions, we used a quantitative approach, and all the data were analyzed using SPSS software (20.0). Firstly, we ran a correlation analysis and reported the Pearson coefficients. For the fourth hypothesis, we used multiple regression analysis. We employed the Enter method to introduce the prediction model. For the second research question, we used mediation analysis with the bootstrapping procedure (with 5000 re-samples) for assessing indirect effects, which was performed in accordance with [42]. To carry out this procedure, we used Preacher and Hayes mediation script for SPSS.

## 5. Results

Table 3 summarizes the descriptive statistics of the scores on the five variables addressed in this study.

The associations between technology-related knowledge, technology integration practices, self-efficacy beliefs, valuative beliefs about technology use, and beliefs about the limits of technology in the case of children with intellectual disabilities are presented in Table 4.

As shown in the table above, technology integration practices and technology-related knowledge correlate strongly and positively (r(83) = 0.65, *p* < 0.01). A moderate correlation was obtained between practices and self-efficacy beliefs (r(83) = 0.35, *p* < 0.01), while there was only a small correlation between practices and valuative beliefs (r(83) = 0.23, *p* < 0.05). Technology integration practices did not correlate significantly with beliefs about the limits of using technology for the education of children with intellectual disabilities. Based on these preliminary analyses, we can conclude that the first three hypotheses can be accepted, while the fourth one is not supported by the results.

Technology-related knowledge correlates only moderately with self-efficacy beliefs (r(83) = 0.37, *p* < 0.01). An interesting result was the correlation between technology-related knowledge and beliefs about the limits of using technology in the case of children with intellectual disabilities. A negative but low correlation was evidenced in our sample based on the used measures (r(70) = −0.294, *p* < 0.05). In other words, the better the score on the technology-related knowledge questions, the lower the score on the beliefs about the limits of using technology for children with disabilities. Teachers either have solutions to these limits or do not treat these beliefs as absolute since new technologies are developing continuously. Valuative beliefs correlate weakly with self-efficacy beliefs.

Further, we ran a multiple linear regression analysis using the Enter method. Besides considering the independent predictive quality of each predictor, as reflected in the values of the Pearson correlation coefficients, we also considered the volume of the sample. We employed an adequate number of participants with respect to the number of predictors (N = 50 + 8 × m, where m is the number of predictors) [43]. In the first block of predictors, we introduced technology-related knowledge based on the strength of correlation, while self-efficacy beliefs were introduced in the second block and valuative beliefs were introduced in the third block. Firstly, we verified whether the basic assumptions for the multiple regression analysis were met. We examined the skewness and kurtosis values, which are indicative of normal distribution. The values of skewness are 0.138, 0.024, −0.189, and −0.978. The kurtosis values range from −0.747 to 2.547. According to [44], these values are considered acceptable for a normal distribution. Tolerance for the predictors ranged from 0.816 to 1. These values are adequate. The VIF values of the predictors that were entered into the model range from 1 to 1.22. These values indicate that there was no multicollinearity. Mahalanobis, D-Cook, and Standardized DfFit did not indicate extreme cases. In addition, the result of the Durbin-Watson test was 1.825, indicating that our data respect the error independence condition.

The results of the regression analysis are presented in Table 5.

The results of the multiple regression analysis support a prediction model with only technology-related knowledge. Based on the value of F and the significance values, the results of all three models are stronger than a random prediction. However, the significance of the change effect with the insertion of self-efficacy beliefs and valuative beliefs is null. The results of the multiple regression analysis provide support for the first hypothesis, according to which technology-related knowledge predicts technology integration practices. However, the results of the regression analysis do not support the fifth hypothesis, according to which factors beyond teachers’ knowledge, self-efficacy beliefs, and valuative beliefs continue to predict teachers’ use of technology. The only significant predictor of technology integration practices was technology-related knowledge. It predicted 42% of the variance in technological integration practices. Teachers that have specific and rich knowledge about technology that can be used in education, math education, and to teach children with disabilities also tend to report greater use of technology in the classroom. 

### 5.1. Mediation Analysis

Based on the correlation pattern and the regression analysis’s results, technology-related knowledge seems to be a mediator of the relationship between self-efficacy and technology integration practices. We conducted a mediation analysis, wherein self-efficacy beliefs were incorporated as an independent variable and technology integration practices were incorporated as the dependent variable. The results indicate that technology-related knowledge (indirect effect = 0.6051; 95% CI = 0.2277 to 0.9964) mediated the effect of self-efficacy on technology integration practices. A diagram of the indirect effects is presented in Figure 1.

### 5.2. Qualitative Analysis of the Answers to the Questionnaire

#### 5.2.1. Technology Related Knowledge

The teachers were asked a question concerning which technology they would recommend for use in the math education of children with intellectual disabilities. The most common response was Wordwall, which appears 35 times out of the 59 responses given, while 11 responses indicated LearningApps. Most responses indicate apps, platforms, tutorials, and educational software without providing specific names. References to platforms with educational resources predominate. Livresq or Livresq library are also mentioned three times in total, while Twinkl is mentioned two times. Kahoot and Genially are each mentioned once. Two respondents also selected Paint and one selected Googleforms. Only two answers refer to specific mathematical digital technologies: Geogebra and Mquest.

The results concerning the item seeking to enumerate the advantages of using mobile math games for the math education of children with intellectual disabilities are presented further.

Of the 68 responses given, a common theme concerns technologies’ effects on the attention of children with ID. A total of 30 separate responses indicate a positive effect on attention. Other frequent responses concerned interactivity, effects on motivation, the appeal of mobile games, etc., (for a complete description of frequent themes and their frequencies, see Appendix A).

Furthermore, teachers were asked to enumerate some advantages of using Interactive Power Point for the math education of children with intellectual disabilities. A total of 77 teachers answered this item. The most frequent responses indicate positive effects on attention, appeal, interactivity, visual support, and the ability to synthesize information.

Questions 4a, 4b, 4c, and 4d requested teachers to give examples of digital technologies that support the development of mathematical representations, discovery learning, learning individualization, and exercise. An answer scored with three points had to be specific. The highest frequency of specific answers for the particular uses of digital technologies in math education was that reported for exercise, while the least was for discovery learning and learning individualization.

Question 5 solicited a yes/no answer as to whether the participant was aware of augmented reality technologies. Only 22.9% answered that they knew of such a technology.

Question 6 asks for examples of apps/software that support the development of virtual recapitulative panels for math learning, while question 7 addresses apps that can be used to make a video tutorial on how to use a computation algorithm, for which the specific answers are 12.2% and 9.6%, respectively.

Question 8 asks the participants to match four resources to four apps. A total of 30.1% correctly matched at least three out of four resources.

Based on these results, we can confidently say that the technology-related knowledge levels in the case of our sample are rather general.

#### 5.2.2. Technology Integration Practices (Practice1–Practice10 (P1–P10))

P1. Which types of digital technologies do you use in math activities?

A high percentage (68.7%) of the respondents indicated Wordwall and LearningApps, while 61.4% indicated educational offline software, 48.2% use apps on a tablet, 43.4% employ general Office software, 24.1% use mobile apps, 38.6% use interactive activities on a smartboard, 20.5% use e-learning platforms, 12.5% use augmented reality, and 3.6% answered this question with the response other.

P2. What would be the average percentage of time that you use digital technology in a lesson?

More than 50% of answers indicated using technology between 10 and 25% of a lesson’s time, while 20% used technology less than 10% of the time.

P3. What would be the mean percentage of a lesson’s content taught with digital technologies? Most of the responses indicated that teachers teach between 30–50% of content using digital technologies.

P4 asked the participants to indicate the digital technologies they used during online school, to which 73.2% responded online activities with Wordwall and LearningApps; 64.4% general Office software; 61% media tutorials, 46.3% apps on a tablet; 34.1% mobile apps; 43.9% e-learning platforms; and 15.9% other.

P5 asked the participants to select which technologies they used once they returned from online school to onsite school. Online activities created on Wordwall and LearningApps represented 67.5% of the responses. A total of 47% of the respondents indicated that they used apps on a tablet. Other frequently used products consisted of educational software. A total of 44.6% also chose general Office programs, 37.3% chose media tutorials, 33.7% selected interactive activities on a smartboard, 25.3% selected e-learning platforms, and selected 21.7% mobile apps. Augmented reality was chosen by 13.3% (N = 11). A total of 10.8% answered other.

P6 asked the participants to give examples of digital technologies they used to teach different types of math content to children with intellectual disabilities, including math prerequisites (cognitive operations, counting, quantitative comparisons, etc.), number concepts and numeracy, basic math computations (addition, subtraction, multiplication, and division), word problem solving (simple, complex, and specific), measurement units, and geometry. For each form of content, the answers were scored with 1 for a class of digital technologies, 2 if they indicated software or an app, and 3 if they indicated technology that was specific to math. The total score ranged from 0–18. A total of 16.8% had a score equal to or more than 12.

P7 asks about the main teaching function for which digital technology was used. A total of 31.3% of the respondents answered that they use it for only one function (either for an exercise, to teach new math concepts, or for assessment), while 43.4% answered that they use it for two functions and 19.3% that they use it for all three functions. For this question, 80 answers were provided. A total of 88.8% reported using digital technology for exercises, 45% reported using it to teach new content, and 47.5% reported using it for assessment.

P8 asked the participants to select the digital resources they use most frequently in math activities. We scored this item as follows: digital resources are never used—0; one frequently used resource—1; two frequently used resources—2; three frequently used resources—3; and four frequently used resources—4.

A total of 45.8% of the participants indicated that they used two resources most frequently. A total of 24.1% indicated that they frequently used three different resources, while only 10.8 indicated that they frequently used four digital resources.

P9 asked the participants about the types of emergent technologies they have used. A total of 78 answers were provided. A total of 33 teachers (42.3%) responded that they do not use such technologies. A total of 3.6% indicated that they had used one emergent technology, and only 1.2 % indicated that they had used two emergent technologies. A total of 11.5% indicated that they had used augmented reality, 26.9% indicated that they had used virtual reality, 5.1% indicated that they had used robotics, and 23.1% responded with other.

P10 asked the teachers to indicate the educational resources they have created, namely, electronic presentations, virtual panels, videos, animation and animated strips, collaborative documents, conceptual maps, educational games or interactive exercises, and assessment resources, using existing apps. They received one point if they indicated between one and two resources, two points for three to four resources, and three points for five to seven digital resources. Consequently, 49.4% and 37.3% indicated that they had created between one to two resources and three to four resources, respectively. A total of 80 answers were provided for this question, of which 77.5% (N = 62) indicated that the participants had created games and interactive exercises; 61.3% (N = 49) that they had created electronic presentations; 47.5% (N = 38) that they had created assessment resources; 18.8% that they had created videos, animations, and/or animated strips; 17.5% that they had created collaborative documents; 25% that they had created conceptual maps, and 11.3% that they had created virtual panels.

### 5.3. Beliefs about Limits of Technology Use for the Education of Children with Intellectual Disabilities

The results are shown in Table 6.

### 5.4. Value Beliefs of Technology in Education

The results are presented in Table 7.

High percentages of teachers indicated that they believed technology is useful and can be integrated at any moment of a lesson, in any type of lesson, and with any type of content. Fewer believed that it is accessible to learners affected by an intellectual disability at any degree of severity and/or is superior to classical assessment and progress-monitoring tools.

### 5.5. Self-Efficacy Beliefs

Results on the self- efficacy beliefs are presented in Table 8.

Regarding the neutral (undecisive) responses, the percentages are rather high. This may indicate the difficulty teachers have in self-assessing their technology integration competencies, which, in turn, impacts their self-efficacy beliefs.

## 6. Conclusions and Discussion

Our study aimed to explore teachers’ technology-related knowledge, technology integration practices, valuative beliefs regarding technology, self-efficacy beliefs about technology use, and beliefs about the limits of technology use in the case of teaching math to children with disabilities. Another aim was to investigate the relationships between these factors affecting technology integration practices in order to identify those that are most important and, therefore, more relevant for changing technology integration levels.

We employed qualitative and quantitative analyses for the questions used to address technology-related knowledge. Based on these analyses, we can conclude that the sample’s technology-related knowledge is rather general (less than 25% of answers are specific to math in general), lacking the integration of all three dimensions from the TPACK framework (technological, pedagogical, and content). This result is in line with the research conducted by [36] as only few of the teachers’ reports in their study demonstrated integrated TPACK knowledge. This may explain why teachers more frequently use educational apps with libraries or e-learning platforms than other available digital technologies. In addition, the relationship may be seen as biunivocal such that by using e-learning platforms and resource libraries with activities, teachers are not as motivated to research and integrate new technologies into their practice. When analyzing the content of several items requiring teachers to enumerate the advantages of using technology to teach math, we observed several themes similar to those related by [36] individualization, motivation and engagement, attention, and various representations. In addition, some themes that were extracted in the aforementioned study were not as present among the answers, such as benefits for life skills and formative assessment.

In order to examine technology integration practices, we also combined qualitative and quantitative data. In practice, teachers prefer apps that have resource libraries, which allow them to individualize their math activities. They mostly use digital technologies for exercise. Most of them report using technology for one or two teaching functions. Most of them do not use specific technology for math but rather more general, subject-free digital technologies. Teachers have the necessary technological knowledge to use such technologies. The resources they use are not very diverse. A high percentage do not use emergent technologies. Most resources they create are interactive exercises, digital presentations, and conceptual maps. They create far fewer resources that allow for collaboration and consist of virtual demonstrative panels. However, this may be due to the fact that interactive exercises and digital presentations require less time to be developed, while video tutorials or animated presentations require more time to plan, implement, and even be mastered by teachers. Ref. [34] provided evidence in support of the impact of time on the use of technology in the case of special education teachers. In addition, another explanation may be related to the didactical value teachers attribute to different resources.

Only a small percentage of teachers have beliefs about the limits of using technologies to teach children with intellectual disabilities. This result may be explained by desirability responses or increased experience with children with intellectual disabilities using at least some devices at a basic level. In addition, the fact that only a small percentage agreed with the statement that digital technologies do not offer children with intellectual disabilities the concrete experiences they need to learn is partly explained by the fact that many consider apps or software as interactive and/or visually supportive media (see the above description of the qualitative analysis on the advantages of using mobile games and interactive PowerPoint presentations for math education). Many use semi-concrete representations that allow for interactivity and the ability to act on these representations, offering visual support for the quantitative information but also for mathematical transformation. However, few examples of specific math-related digital technologies for supporting mathematical representations were given. Two specific answers referred to virtual manipulatives and numerical axes. Some answers also indicated the use of interactive apps that allow for the grouping of objects and operations with groups of objects.

Beliefs about the value of technology in the classroom can be grouped into beliefs about the use of technology in teaching a lesson, beliefs about the accessibility of technologies for children with any level of intellectual deficit, and beliefs about the use of technology for assessment and progress monitoring. Most teachers perceived that technology can be used for teaching, while fewer found that it can be used for assessment and progress monitoring.

Most teachers strongly agreed with the self-efficacy statements. However, an elevated percentage responded indecisively, which may indicate difficulties in the self-assessment of the competencies of technology integration. In our quantitative analysis, we obtained a correlation between self-efficacy beliefs and technology integration practices, which is in line with the findings of [34]; however, the strength of the relationship is lower. In addition, [36] evidence for a relationship between self-efficacy in using technology and the use of technology in the classroom.

When introduced in the regression model together with technology-related knowledge, self-efficacy does not contribute a significant change. However, in our study, self-efficacy correlates with technology-related knowledge in a manner very similar to technology integration practices. The correlation with technology-related knowledge is moderate, indicating that the two constructs are rather independent as measured using the set of questions that we selected. Taimalu and Luik [13] found a strong direct effect of perceived knowledge of technology and its integration and self-efficacy beliefs on the use of technology. Our findings indicate that the more teachers improve their technology-related knowledge, the higher their self-efficacy with respect to using technology. This confidence in using technology is more strongly associated with knowledge they have gained rather than beliefs about its value. We further analyzed the relationships between technology-related knowledge, self-efficacy, and technology integration practices through a mediation analysis. The results indicate a significant indirect effect of self-efficacy on technology integration practices via technology-related knowledge.

Our regression analysis evidences the strong predictive value of technology-related knowledge with respect to technology integration practices in math education. This result is in line with the findings of [11,12,13]. However, beliefs lost their predictive power when introduced along with technology-related knowledge. Technology-related knowledge that is specific is important not only for technology integration but also for beliefs about its value and confidence in the use of technology. This is an important factor to address when aiming to improve technology integration in special education math classes.

The results of this study are relevant to special education teachers’ initial formation and for designing professional development programs on the topic of technology integration in mathematics special education. Technology integration can be strongly predicted according to teachers’ technology-related knowledge. This knowledge is dynamic; it organizes itself and is constantly increasing and refining. Teachers require knowledge on the affordances and limitations of technology for use in teaching math but also on how to make proper selections of technology for specific learning needs; how to adapt, if necessary, learning sequences using technology for a meaningful conceptual learning experience; and how to enable the acquisition and mastery of math operations in a manner tailored to the individual characteristics of children with intellectual disabilities. In addition, teachers need support in organizing this integrated knowledge. Particularly with regard to math in special education, there is a need to integrate specific math content knowledge with pedagogical knowledge, technological knowledge, and psycho-pedagogical knowledge on the characteristics of learning and understanding math concepts among students with intellectual disabilities. Most digital technologies are created for children with typical development; thus, teachers need to adapt and combine various teaching strategies and use an adapted concrete-representational-abstract approach in order to adequately form mathematical representations, math concepts, etc. Some technologies do not offer many alternatives to adapt content in terms of visual representation, speed, and the type of representation. There is currently an abundance of digital technologies designed for math education. It is recommended to place particular emphasis on the curriculum of teachers formation and professional development’ programs. In addition, given the abundance of such technologies, teachers may be supported through teaching frameworks/approaches that are more specified. Curricular content of professional development programs that is more specific to the teaching subject is recommended [36,45].

Another practical implication of the current study is related to the finding that self-efficacy beliefs predict technology integration in the math education class through the effect on technology- related knowledge. Self-efficacious teachers have the confidence required to gather and organize knowledge on technology for math education and then apply it. Previous work [46] recommended including ICT problem-solving skills into the curriculum, which would serve to strengthen teachers’ self-efficacy beliefs and contribute to knowledge refinement and organization.

## 7. Limitations

One limitation of the current study was imposed by the instrument we used to collect data. Even though the dimensions have good reliability coefficients and content validity was addressed in the procedure, more information on the validity of the questionnaire would have increased the robustness of the results and the value of the conclusions. However, this was considered a valuable alternative to self-reported TPACK knowledge. A similar approach, although based on an interview and not a questionnaire, addressed TPACK knowledge [36] On the other hand, standardized measures of self-efficacy with respect to technology are mostly technical [46] and general; for this reason, we contextualized items while referencing to such instruments. Another limitation concerns the inability to unveil the rich belief-related content relevant to technology use among teachers that work in special education. We find that more research is needed as greater experience with technologies represents a great source for the development of some mental models on what works and why as well as what does not. In addition, since beliefs are dynamic and changeable, it is possible that our results are limited to the present moment and cannot predict future behaviors. For a more realistic view of the impact of beliefs on technology integration practices, more measures analyzed at different times should be considered. In addition, we did not integrate pedagogical beliefs that could weigh the beliefs about the limits of technology and whether an approach is student-centered or teacher-centered.

In conclusion, our results support findings not only from research on general education teachers but also on special education teachers. Our research also provides new insight on the relationship between self-efficacy and technology integration practices, unveiling that self-efficacy should be linked to self-assessment competencies in using technology in the classroom, especially when self-efficacy tools are contextualized. In addition, we were able to evidence the practice of technology integration after the occurrence of a normative event, such as in a lockdown provoking online schooling. We showed that teachers use many digital technologies for many functions and various types of content and for a third of the teaching time. In addition, we were able to underline some needs concerning practices. One such need is the development of diversity in terms of the functions and types of technology frequently employed and the further integration of emergent technologies as these hold great potential for a more active involvement of students, thereby enabling deeper immersion into a teaching subject.

## Figures and Tables

**Figure 1 children-10-00918-f001:**
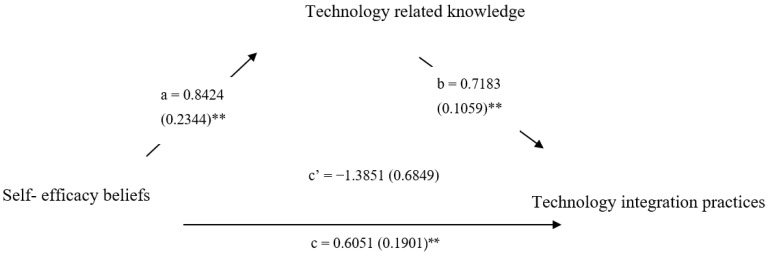
Mediation diagram of the indirect effects. Unstandardized regression weights and standard errors (in parentheses); path a = independent variable (IV) to mediator; path b = mediator to dependent variable (DV); path c = total effect of IV on DV; c’ = direct effect of IV on DV; ** *p* = 0.01.

**Table 1 children-10-00918-t001:** Sections of the questionnaire and the number of items.

Section	Number of Items
Informed consent and demographic information, including the general information on the number of children taught by a teacher, the severity of the deficit, and the gap in math competencies of the learners	17 items
Technology-related knowledge	11 items
Technology integration practices	10 items
Valuative beliefs about the use of technology	6 items
Beliefs about the limits of using technology for the education of children with intellectual disabilities	3 items
Self-efficacy beliefs	5 items

**Table 2 children-10-00918-t002:** Reliability coefficients.

Measure	No. of Items	α Cronbach
Valuative beliefs about the use of technology	6 items	0.72
Beliefs about the limits of using technology for the education of children with intellectual disabilities	3 items	0.62
Self-efficacy beliefs	5 items	0.79
Technology-related knowledge	11 items	0.81
Technology integration practices	10 items	0.49

**Table 3 children-10-00918-t003:** Descriptive statistics.

Variables	N	Min	Max	M	SD	SE
Technology-related knowledge	83	1	26	13.55	5.71	0.63
Technology integration practices	83	7	41	22.43	6.74	0.74
Self-efficacy	83	9	25	18.65	2.51	0.27
Valuative beliefs about the use of technology	83	0	6	2.72	1.73	0.19
Beliefs about limits of technology for children with intellectual disabilities	70	0	3	0.61	0.92	0.11

**Table 4 children-10-00918-t004:** Correlation matrix.

	1	2	3	4
Technology integration practices				
Technology-related knowledge	0.656 **			
Self-efficacy	0.354 **	0.371 **		
Beliefs regarding technology’s limits	−0.017	−0.294 *	−0.193	
Valuative beliefs	0.230 *	0.140	0.265 *	−0.218

** Correlation is significant at the 0.01 level (2-tailed). * Correlation is significant at the 0.05 level (2-tailed).

**Table 5 children-10-00918-t005:** Regression analysis results.

Model	Variables	B	SE	β	t	R	Adj R^2^	F
1.	(Constant)	11.936	1.455		8.20 **			
	TRK	0.775	0.099	0.656	7.82 **	0.656	0.423	61.211 **
2.	(Constant)	6.277	4.21		1.49	0.667	0.431	32.025 **
	TRK	0.718	0.106	0.608	6.78 **			
	SE	0.344	0.241	0.128	1.431			
3.	(Constant)	6.584	4.19		1.57			
	TRK	0.712	0.105	0.603	6.75 **			
	SE	0.265	0.246	0.099	1.077	0.677	0.437	22.248 **
	VB	0.466	0.335	0.120	1.393			

TRK—technology-related knowledge, SE—self-efficacy beliefs, and VB—valuative beliefs; ** t is significant at the 0.01 level; dependent variable—technology integration practices.

**Table 6 children-10-00918-t006:** Percentage of teachers that reported beliefs about the limits of technology use for the education of children with intellectual disabilities.

Beliefs	Number of Respondents that Agreed with It	Percentage
Digital technologies cannot be implemented in the education of children with intellectual disabilities	12	14.5%
Children with moderate and severe intellectual disabilities do not have the minimum digital abilities required to use digital technologies efficiently.	16	19.3%
Children with intellectual disabilities need concrete experiences to learn and digital technologies do not confer these.	18	21.7%

**Table 7 children-10-00918-t007:** Percentages of respondents that reported holding valuative beliefs about technology.

	DT at Any Moment of a Lesson	DT in Any Type of Lesson	DT for Any Type of Content	There Are DT Accessible at Any Degree of Severity	DT Better for Assessment than Classic Paper and Pencil	DT Better to Monitor Students’ Progress
N	63	60	42	24	18	19
%	75.9%	72.3	50.6%	28.9%	21.7%	22.9%

DT—digital technologies.

**Table 8 children-10-00918-t008:** Percentages of answers regarding self-efficacy beliefs.

Answer	SE1 (%)	SE2 (%)	SE3 (%)	SE4 (%)	SE5 (%)
Strongly disagree	1.2	-	-	1.2	2.4
Disagree	2.4	1.2	9.6	2.4	7.2
Neutral	19.3	9.6	34.9	19.3	25.3
Agree	69.9	81.9	51.8	69.9	56.6
Strongly agree	7.2	7.2	3.6	7.2	8.4

SE—self efficacy beliefs; SE1—my lessons appropriately combine educational approaches with technology and mathematical content; SE2—I can select technologies that enhance the content of teaching and improve the methods used and the performance of my students; SE3—I am well aware of the limitations that different digital technologies have with respect to the learning of mathematics through different forms of organization and teaching methods among students with intellectual disabilities; SE4—I can select digital technologies that can be easily used by my students with intellectual disabilities to create meaningful learning experiences regarding mathematical operations; SE5—I continuously update my knowledge of new digital technologies for the mathematics education of students with disabilities either through self-study or by attending training courses, workshops, etc.

## Data Availability

The data presented in the study are available from the corresponding author upon request. The data are not publicly available due to privacy and ethical considerations.

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
