# Peer review of "Factors That Support Technology Integration Practices in the Math Education of Children with Intellectual Disabilities"

_children, 2023, doi:10.3390/children10060918_

Round 1
Reviewer 1 Report
This is a well written article, with a carefully applied and analyzed survey methodology that addresses an important research question as, indeed, there is very limited research on integrating technologies in math education of children with special educational needs. I especially appreciate the efforts that authors did to customize the original questionnaire contents and create an ad hoc version to address the competencies of the special education curriculum as this addresses an important research gap. As follows, authors will find quite a few comments that, although they are minor and I believe can be easily addressed, I hope they can contribute to the improvement of the manuscript.
- 3. Purpose: Math education factor is included as a research question and so it would be recommended to also mention it earlier in the general purpose of the study (as it is also included in the title of the article as an important matter).
- 4. Participants:
o I would encourage the authors to move this section to the Methods section as, generally, the introduction section ends with the purpose, research questions and hypothesis, and the method section starts with the participants description.
o When authors indicate that “A percentage of 66.3 work with 181 students that present mild and moderate intellectual disabilities, while 51.8% work with 182 students with profound, severe and associated”, it would be advised to add some reference to gently contextualize these clinical diagnostic labels (e.g., DSM-V-TR, ICD-11).
o Could you describe what a “resource teacher” is? It may have different names/meanings across countries and cultures (e.g., instructional resource teacher, special education teacher, one-to-one or small group support teacher).
- 5.1. Procedure: it would be desirable, if this is possible, to know when the questionnaire was administered (start and end dates). Also, knowing how the link to the Google form questionnaire was sent, would be convenient (e.g., via email).
- 5.2.1 A few things:
o After this sentence “an additional section that was not included in this research asked about perceived barriers and facilitators of technology integration into the classroom”, it would be needed an explanation of why this last part has not been included in this article. If, for instance, it is because there will be another article with other research questions focusing on barriers and facilitators, this would help the readers to follow your ongoing work.
o Also, review this sentence: “All answers on the open-ended questions *had to rated* based on clear criteria” (had to be rated, were rated, …).
o What are the clear criteria? (line 236). Providing them (maybe as supplementary material?), would facilitate the replication of the study.
- 6.2.2. In P1-P10, what does P mean? Practice? Please, indicate it.
- Table 6: review this sentence, “Children with moderate and severe intellectual disabilities do not have minimal digital abilities *to in order use* digital technologies efficiently” (in order to use?)
- Table 8: “SE1, SE2, …” I guess these represent the 5 items of the self-efficacy beliefs scale but lacking the information of what they are, it does not facilitate the understanding of the results.
- Last conclusion: as in the purpose, I miss here to find words related to the matter of mathematics education. Is it really core in the article or just a part of it? I encourage the authors to discuss this and make the best decision (adapting either the indicated parts or the title and abstract accordingly if need be).
Author Response
Reviewer 1
This is a well written article, with a carefully applied and analyzed survey methodology that addresses an important research question as, indeed, there is very limited research on integrating technologies in math education of children with special educational needs. I especially appreciate the efforts that authors did to customize the original questionnaire contents and create an ad hoc version to address the competencies of the special education curriculum as this addresses an important research gap.
Thank you for your appreciation of the paper.
Comment 1: Purpose: Math education factor is included as a research question and so it would be recommended to also mention it earlier in the general purpose of the study (as it is also included in the title of the article as an important matter).
Thank you for your suggestion, we added the following text in the revised manuscript:
The purpose of the current study is to investigate special education teachers’ technology- related knowledge, self-efficacy beliefs, beliefs about the use of technology for education and specific, in the case of children with intellectual disabilities, as well as practices of integrating technology into the classroom, in Math education.
Comment 2: Participants: I would encourage the authors to move this section to the Methods section as, generally, the introduction section ends with the purpose, research questions and hypothesis, and the method section starts with the participants description.
We have moved the Participants under the Methods section and renumbered the sections.
Comment 3: When authors indicate that “A percentage of 66.3 work with 181 students that present mild and moderate intellectual disabilities, while 51.8% work with 182 students with profound, severe and associated”, it would be advised to add some reference to gently contextualize these clinical diagnostic labels (e.g., DSM-V-TR, ICD-11).
Thank you for your recommendation. We added World Health Organization, 2022 as reference.
Comment 4: Could you describe what a “resource teacher” is? It may have different names/meanings across countries and cultures (e.g., instructional resource teacher, special education teacher, one-to-one or small group support teacher).
We added in the revised manuscript the definition for resource teachers (teachers that offer support to children in inclusive schools)
Comment 5: Procedure: it would be desirable, if this is possible, to know when the questionnaire was administered (start and end dates). Also, knowing how the link to the Google form questionnaire was sent, would be convenient (e.g., via email).
The questionnaires were sent via email, between May and August 2022.
Comment 6: After this sentence “an additional section that was not included in this research asked about perceived barriers and facilitators of technology integration into the classroom”, it would be needed an explanation of why this last part has not been included in this article. If, for instance, it is because there will be another article with other research questions focusing on barriers and facilitators, this would help the readers to follow your ongoing work.
Thanks for your suggestion. We have included the following paragraph in the revised manuscript: This part was purposely left out to be included in another article focusing on barriers and facilitators in technology integration.
Comment 7 Also, review this sentence: “All answers on the open-ended questions *had to rated* based on clear criteria” (had to be rated, were rated, …)
We corrected this in the revised manuscript.
Comment 8 What are the clear criteria? (Line 236). Providing them (maybe as supplementary material?), would facilitate the replication of the study.
We have included this information in the supplementary material as indicated.
Comment 9: 6.2.2. In P1-P10, what does P mean? Practice? Please, indicate it.
We have added this information in the revised manuscript. It means Practice1–Practice10, P1- P10
Comment 10: Table 6: review this sentence, “Children with moderate and severe intellectual disabilities do not have minimal digital abilities *to in order use* digital technologies efficiently” (in order to use?)
Thank you for your suggestion, we have corrected this.
Comment 11: Table 8: “SE1, SE2, …” I guess these represent the 5 items of the self-efficacy beliefs scale but lacking the information of what they are, it does not facilitate the understanding of the results.
We have added this information.
Comment 12: Last conclusion: as in the purpose, I miss here to find words related to the matter of mathematics education. Is it really core in the article or just a part of it? I encourage the authors to discuss this and make the best decision (adapting either the indicated parts or the title and abstract accordingly if need be).
Integrating technologies into the Math education is a core topic, indeed. We added information the following related to this matter.
Teachers need knowledge on affordances and limitations of technology for use in teaching Math, but also on how to make proper selections of technology for specific learning needs, how to adapt, if necessary, learning sequences with technology for a meaningful conceptual learning, Math operations acquisition and mastery to the individual characteristics of children with intellectual disabilities.
Also, we added more implications related to Math education.
Reviewer 2 Report
In this study, the authors administered an online questionnaire to 83 teachers to study the relationship between several variables: technology related knowledge, technology integration practices, value beliefs about the use of technology, beliefs about the limits of using technology in the education of children with intellectual disabilities, self- efficacy beliefs.
Although the topic is interesting and has potential applicative implications, I find that the study has methodological limitations that do not allow its publication in its current state.
From what I understand, the questionnaire used was adapted from various instruments available in the literature and largely developed by the authors themselves. An evaluation of the psychometric properties (beyond internal consistency) of the instrument is completely lacking (e.g., factor structure and measurement invariance, concurrent, convergent, and predictive validity, test–retest reliability). This is fundamental for any new questionnaire. Without this step, any conclusions drawn risk being misleading. Consequently, I do not feel able to review results and discussion.
The literature review can be improved. See https://www.sciencedirect.com/science/article/pii/S0360131520301512 for references of potential relevance.
English writing would benefit from a general revision.
Author Response
Reviewer 2:
Comments and Suggestions for Authors
In this study, the authors administered an online questionnaire to 83 teachers to study the relationship between several variables: technology related knowledge, technology integration practices, value beliefs about the use of technology, beliefs about the limits of using technology in the education of children with intellectual disabilities, self- efficacy beliefs.
Comment 1: Although the topic is interesting and has potential applicative implications, I find that the study has methodological limitations that do not allow its publication in its current state.From what I understand, the questionnaire used was adapted from various instruments available in the literature and largely developed by the authors themselves. An evaluation of the psychometric properties (beyond internal consistency) of the instrument is completely lacking (e.g., factor structure and measurement invariance, concurrent, convergent, and predictive validity, test–retest reliability). This is fundamental for any new questionnaire. Without this step, any conclusions drawn risk being misleading. Consequently, I do not feel able to review results and discussion.
Thank you for raising this issue.
In the initial manuscript, we conducted an internal consistency analysis, and we used a checklist to address the content of the questionnaire. It was filled in by 3 independent experts, in order to further address the theoretical expected structure. However, given the limited statistical power, since we had 83 participants and 55 items, we did not conduct a factorial analysis. We addressed this issue in the limits section, as follows:
More information on the validity of the questionnaire would have increased the robustness of the results and the value of the conclusions.).
Comment 2: The literature review can be improved. See https://www.sciencedirect.com/science/article/pii/S0360131520301512 for references of potential relevance.
We added a paragraph related to Math interventions in the case of children with special needs.
Knowledge on evidence- based practices in using technology in Math education or Math intervention for children with special needs is building up. Spooner, Root, Saunders, & Browder (2018) indicated the efficacy of using manipulatives and technology- aided instruction. Benavides- Varela, Callegher, Fagiolini, Leo, Altoè, & Lucangeli (2020) indicated a moderate effect of interventions based on digital strategies for children with specific learning disorders. Also, they underline the high variability across studies and the need to look further into factors that relate to intervention efficacy. This scarce research, as well as the variability in research results may contribute to the limited response of teachers to updating knowledge.
Comment 3: English writing would benefit from a general revision.
Thank you for your suggestions. We made a general revision.
Reviewer 3 Report
Thanks for the opportunity to review the paper Factors that support technology integration practices in the Math education of children with intellectual disabilities. I recommend this be published after some minor revision.
Overall, the manuscript is well-written, and the methodology sounds fine. I only have some concern on the methodology and a few suggestions - as I documented below - to further strengthen argumentations and streamline the text.
First of all, how do the authors think their instrument is different from other similar instruments already present in the literature (see, for example, Dong et al., 2020): what is the novelty? what is the contribution to the scientific community?
Dong, Y., Xu, C., Chai, C. S., & Zhai, X. (2020). Exploring the structural relationship among teachers’ technostress, technological pedagogical content knowledge (TPACK), computer self-efficacy and school support. The Asia-Pacific Education Researcher, 29(2), 147-157.
Also, I reckon that this literature cannot be discussed if not in the light of literature on teachers' acceptance of ICT, but also techno-pessimism and techno-stress. I suggest, in the introduction/literature review, to better justify your hypothesis by commenting on some few recent investigations on these factors: e.g.,
Monacis, L., Limone, P., Ceglie, F., Tanucci, G., & Sinatra, M. (2019). Exploring individual differences among teachers' ICT acceptance: A path model and role of experience. Human technology, 15(2), 279.
Sulla, F., Ragni, B., Miriana, D. A., & Rollo, D. (2022). Teachers’ emotions, technostress, and burnout in distance learning during the COVID-19 pandemic. In Proceedings of Third Workshop of Technology Enhanced Learning Environments for Blended Education–The Italian e-Learning Conference 2022 (Vol. 3265, pp. 1-13). CEUR.
Tomczyk, Ł., Jáuregui, V.C., de La Higuera Amato, C.A. et al. Are teachers techno-optimists or techno-pessimists? A pilot comparative among teachers in Bolivia, Brazil, the Dominican Republic, Ecuador, Finland, Poland, Turkey, and Uruguay. Educ Inf Technol 26, 2715–2741 (2021). https://doi.org/10.1007/s10639-020-10380-4
Methods
Questionnaire development, line 197
I am not sure if I get this correctly. I wonder why the authors say that “self-efficacy beliefs section was developed after Schmidt et al (2009) - this one in particular does not seem to address self-efficacy beliefs though, but only TPAK - and others” and then, on line 210, they claim they did not use a standardized measure for it?
In any case, it is necessary to justify this sentence by citing previous work: “Many instruments used to assess technological pedagogical content knowledge are evidenced to rather address self-efficacy.” Moreover, I would discuss the decision not to use standardized measure of self-efficacy in the limits section.
For example, Dong et al (2020) - aforementioned - in a study with similar scopes as yours use a measure of computer self efficacy (which I think fit the aim of investigations as yours) and the items comes from two standardized measures.
Line 214
I would like to know more on some of the phases listed here. For example, how did you decided whether the experts were qualified? did they agreed upon everything? did the authors check for content validity using some scale? If so, can you provide Items and scale CVIs? otherwise, I suggest to address this in limits section. In general, I would spend at least one sentence per each bullet point that explains the task in more details.
Line 236
‘clear criteria’: I would explain right here what the criteria was.
Results
Line 291
A normally distributed data has both skewness and kurtosis equal to zero. It is near-normal if skewness and kurtosis both ranges from -1 to 1. I suggest to do a Kolmogorov–Smirnov test.
Table 5
Personally, I’d like to see the constant as well because then readers can construct the full regression model if they need to. Moreover, in footstep notes I would add p value that correspond to **, and also what is the dependent variable.
Table 6
it would be interesting to know if there are differences in these beliefs based on teachers' age and number of years' teaching. As there are differences based on teachers’ experience for example on their acceptance of technology (e.g., Monacis et al., 2019; aforementioned).
Discussion
Within the conclusion I suggest to add a paragraph/section where the authors discuss possible implications for practice both of teachers, especially special needs teachers, and of professionals that the authors wishes could benefit of the use of the developed instrument - based on the results of their study.
Author Response
Reviewer 3
Comments and Suggestions for Authors
Thanks for the opportunity to review the paper Factors that support technology integration practices in the Math education of children with intellectual disabilities. I recommend this be published after some minor revision.
Overall, the manuscript is well-written, and the methodology sounds fine.
Thank you for you appreciation of the paper!
I only have some concern on the methodology and a few suggestions - as I documented below - to further strengthen argumentations and streamline the text.
Comment 1:First of all, how do the authors think their instrument is different from other similar instruments already present in the literature (see, for example, Dong et al., 2020): what is the novelty? what is the contribution to the scientific community?
This instrument has been developed to address specific technology integrated knowledge in Mathematics learning of children with intellectual disabilities, as many instruments require self-rating of this knowledge. Also, we developed a scale to address technology integration practices, that incorporates more information than the frequency of use. Furthermore, while trying to refine the types of beliefs that play a role in technology integration, we have formulated value beliefs and separated them from beliefs about the limits of technology. The content for these subscales was based on the results of Thurm (2020), Anderson and Putman (2020). Self-efficacy beliefs were formulated having a starting point Schmidt, Baran, Thompson, Mishra, Koehler, & Shin (2009), Survey of preservice teachers’ knowledge of teaching and technology (main source). The content was contextualized to teaching Math with technologies.
Dong, Y., Xu, C., Chai, C. S., & Zhai, X. (2020). Exploring the structural relationship among teachers’ technostress, technological pedagogical content knowledge (TPACK), computer self-efficacy and school support. The Asia-Pacific Education Researcher, 29(2), 147-157.
- For the Technology- related knowledge scale, we used open-ended questions and a matching item to address the technology related knowledge of teachers. Three questions addressed knowledge on digital technologies used for Math education of children with intellectual disabilities (e.g. Enumerate some advantages of using interactive PowerPoint in the Math education of children with intellectual disabilities), three questions addressed pedagogical uses of technology in Math education (e.g. Give an example of technology that can support the formation of Mathematical representations), two questions addressed knowledge on different types of technologies for learning (e.g. Match the following resources with the apps/ software you can use to create them: Conceptual maps/ Educational movies/ Electronic presentation/ Interactive activities for practice/ Prezi/ Storyjumper/ Mindmeister/ Twinkl.).
- For the Technology integration practices, we used questions with response alternatives. All questions address what they used or created with digital technologies. “When returning from online school to on- site, I have used the following technologies: Educational software/ Office software/Mobile apps/Tablet apps/Online activities in Wordwall, LearningApps/Interactive activities on the smartboard/ e-learning platforms /Augmented reality/ other”.
- Value beliefs about using technology in the education of children with intellectual disabilities were selected- the content included specific uses of technology for instruction and assessment, type of content, stages of the lesson etc.
- Beliefs about the limits of using technology in the education of children with intellectual disabilities- the content was derived from research that addressed this topic through interviewing special education teachers.
- Self- efficacy beliefs- Though using Schmidt et al. (2009) in order to address different types of integrated knowledge when using technology, we used first item on the technological- pedagogical- content knowledge. Item2 is addressing self- efficacy in using technological pedagogical knowledge in teaching. Items 3 and 4 address self- efficacy in using technology for teaching Math in the case of children with intellectual disabilities. The last item addresses self- efficacy in improving knowledge and skills related to the technology integration in education.
Comment 2: Also, I reckon that this literature cannot be discussed if not in the light of literature on teachers' acceptance of ICT, but also techno-pessimism and techno-stress. I suggest, in the introduction/literature review, to better justify your hypothesis by commenting on some few recent investigations on these factors: e.g., Give an example of technology that can support the formation of Mathematical representations/
Monacis, L., Limone, P., Ceglie, F., Tanucci, G., & Sinatra, M. (2019). Exploring individual differences among teachers' ICT acceptance: A path model and role of experience. Human technology, 15(2), 279.
Sulla, F., Ragni, B., Miriana, D. A., & Rollo, D. (2022). Teachers’ emotions, technostress, and burnout in distance learning during the COVID-19 pandemic. In Proceedings of Third Workshop of Technology Enhanced Learning Environments for Blended Education–The Italian e-Learning Conference 2022 (Vol. 3265, pp. 1-13). CEUR.
Tomczyk, Ł., Jáuregui, V.C., de La Higuera Amato, C.A. et al. Are teachers techno-optimists or techno-pessimists? A pilot comparative among teachers in Bolivia, Brazil, the Dominican Republic, Ecuador, Finland, Poland, Turkey, and Uruguay. Educ Inf Technol 26, 2715–2741 (2021). https://doi.org/10.1007/s10639-020-10380-4
We added a paragraph in order to address this recommendation, as it follows:
Digital technology acceptance is a relevant factor for its integration into the classroom. There are several models for ICT integration, of which the most popular is the technology acceptance model (Davis, 1989; Venkatesh & Davis, 2000). According to this model technology acceptance considers two variables, such as the perceived easiness of using technologies, and the perceived usefulness. There many ways of investigation this subjective perception. For instance, Tomczyk, Szotkowski, Fabiś, Wąsiński, Chudý, & Neumeister (2017) delineated a typology based on the attitudes towards new media: techno-pessimist, techno- optimist, techno- realist, and techno-ignorant. Tomczyk et al., (2021) in a recent paper investigated teachers’ opinions and experiences with the implementation of new technologies in the classroom. Their study involved 873 teachers from 8 different countries, and they had important findings regarding techo-optimism. Even though there are different approaches among countries, when analyzing the data from the survey, in general teachers tend to be techno-optimists. On the other hand, Sulla, Ragni, Angelo and Rollo (2022) in their study mention the fact that the use of information and communication technologies can cause teachers stress (technostress), which can in-fluence their well-being, especially emotional state. Their sample consisted of 1210 teachers, majority of them from Italy, and their findings suggest that teachers may experience negative emotions while practicing distance learning, and that the risk of technostress (with three related domains anxiety, fatigue and ineffectiveness) can predict burnout among teachers. However, we also need to consider personal differences when investigating the use of technology among teachers. The experience, for example can be a great predictor of technology acceptance among teachers as shown by Monacis and his collaborators (2019). Their results show that teachers with 21 years of experience or more seem to be more unlikely to adopt technological tool within their classrooms, compared to teachers less experience.
Comment 3: I am not sure if I get this correctly. I wonder why the authors say that “self-efficacy beliefs section was developed after Schmidt et al (2009) - this one in particular does not seem to address self-efficacy beliefs though, but only TPAK - and others” and then, on line 210, they claim they did not use a standardized measure for it?
The answer on comment 1 it applies to this point.
In any case, it is necessary to justify this sentence by citing previous work: “Many instruments used to assess technological pedagogical content knowledge are evidenced to rather address self-efficacy.”
We added the references.
Moreover, I would discuss the decision not to use standardized measure of self-efficacy in the limits section. /
We added a information about this limit in this section and in the section describing how we developed the questionnaire.
“On the other hand, standardized measures of self- efficacy with technology are mostly technical (Dong et al., 2019) and general, for this reason we contextualized items while referencing to such instruments.”
Many instruments assessing technology self- efficacy are technical (Dong, Xu, Chai, & Zhai, 2019) and general.
For example, Dong et al (2020) - aforementioned - in a study with similar scopes as yours use a measure of computer self efficacy (which I think fit the aim of investigations as yours) and the items comes from two standardized measures.
Comment 4: I would like to know more on some of the phases listed here. For example, how did you decided whether the experts were qualified? did they agreed upon everything? did the authors check for content validity using some scale? If so, can you provide Items and scale CVIs? otherwise, I suggest to address this in limits section. In general, I would spend at least one sentence per each bullet point that explains the task in more details.
We have this limit addressed in the Limits section.
More information on the validity of the questionnaire would have increased the robustness of the results and the value of the conclusions. However, this was considered a valuable alternative to self- reported TPACK knowledge. A similar approach, however based on interview, not a questionnaire, addressed the TPACK knowledge (Anderson & Putman, 2020).
We also provided more info to explain the development task in detail. We included this:
The development of the questionnaire was based on existing works in the field. However, we needed a more contextualized investigation instrument that would allow for the measurement of our variables. Previous indicated work was consulted, and constituted the basis for the development of the items. We used contextual information, and for this reason, we consulted the Special education Math Curriculum. Also, we considered a wide range of digital technologies, in order to obtain good sample. Each item was further analyzed for its content and for its categorization to a particular construct. The items that were not clear, were reformulated, those redundant were eliminated, those that did not fit the construct, were either reformulated or eliminated. Once we decided on the items that remained, we further analyzed the content validity via a multi-step procedure. We provided the experts with a clear definition of the construct. We used an analysis checklist in which we introduced the items on the technology related knowledge and technology integration practices. Experts were asked to indicate the category to which the items belong, also whether the formulation was clear. A good agreement was obtained on most of the items except for Practice 7. We discussed it with the expert and afterwards, he reconsidered the category. One item on the Practices subscale was eliminated, as it was not clear in formulation and it did not receive agreement. The remaining items were included in the final form of the questionnaire.
Comment 5: Line 236 ‘clear criteria’: I would explain right here what the criteria was.
Addressed in supplementary material section.
Comment 6: A normally distributed data has both skewness and kurtosis equal to zero. It is near-normal if skewness and kurtosis both ranges from -1 to 1. I suggest to do a Kolmogorov–Smirnov test.
Response: Indeed, the distribution is near normal. However, we referred to Hair, Black, Babin, & Anderson (2010), indicating that these values are considered acceptable for normal distribution.
Comment 7: Table 5 Personally, I’d like to see the constant as well because then readers can construct the full regression model if they need to. Moreover, in footstep notes I would add p value that correspond to **, and also what is the dependent variable.
We addressed all these points in Table 5.
Comment 8: Table 6 it would be interesting to know if there are differences in these beliefs based on teachers' age and number of years' teaching. As there are differences based on teachers’ experience for example on their acceptance of technology (e.g., Monacis et al., 2019; aforementioned).
While we thank you for these suggestions, we did not have an apriori hypothesis related to this issue and for this reason, we did not conduct the analyses for these differences.
Comment 9: Discussion Within the conclusion I suggest to add a paragraph/section where the authors discuss possible implications for practice both of teachers, especially special needs teachers, and of professionals that the authors wishes could benefit of the use of the developed instrument - based on the results of their study.
Implications for practice
Results of the study are relevant to special education teachers’ initial formation, as well as for designing professional development programs on the topic of technology integration in Mathematics special education. Technology integration is strongly predicted by the technology related knowledge the teachers have. This knowledge is dynamic, it organizes itself, it’s constantly increasing, refining. Teachers need support in organizing this integrated knowledge. In particular for Math in special education, there is a need to integrate specific Math content knowledge with pedagogical knowledge, technology knowledge, and also with psychopedagogical knowledge on characteristics of learning and understanding Math concepts by students with intellectual disabilities. Most digital technologies are created for children with typical development and teachers need to adapt and combine various teaching strategies and use adapted concrete- representational- abstract approach in order to adequately form Mathematical representation, Math concepts etc. . Some technologies do not offer many alternatives to adapt the content, in terms of visual representation, speed, types of representations. There is currently on affluence of digital technologies in Math education. It is recommended to attribute a particular emphasis on the curriculum of the teachers ‘formation and professional development programs. Also, given this affluence of such technologies, teachers may be supported with teaching frameworks/ approaches that are more specified. A curricular content of the professional development program that is more specific to the teaching subject is recommended (Anderson & Putman, 2020, Ciampa, 2017).
Another practical implication of the current study is related to the finding that self- efficacy beliefs predict technology integration in Math education class through the effect on technology related knowledge. Self-efficacious teachers are confident to gather and organize knowledge on technology for Math education and then apply it. Previous work (Dong et al., 2019) recommends to include ICT problem- solving skills into the curriculum, that would strengthen their self- efficacy beliefs, and contribute to knowledge refinement and organization.
Reviewer 4 Report
Thank you for the opportunity to review the manuscript entitled, “Factors that support technology integration practices in the Math education of children with intellectual disabilities. ”
Below I provide my overall impressions followed by more specific comments.
OVERALL COMMENTS:
1. This manuscript draws attention to an important topic in the effectiveness of technology integration and math education among teachers in special education.
2. Parts of this manuscript could be strengthened. For example, include more details on the rationale to develop a questionnaire for the current study. Are there any existing scales for these measures?
3. Please provide more detailed information regarding participants. Other than gender, years in teaching, and qualifications, any socio-demographic information was collected?
4. Please provide more detailed information regarding the sample selection process and exclusion criteria, if any. When was data collection performed and completed?
5. On p. 6, in Table 2, the reliability coefficient of .62 is quite low. Please add an explanation for this low reliability and item content.
6. On p. 8, line 319, please fill in the specific figures in the sentence.
7. Information in Tables 6 through 8 can be condensed into a short paragraph instead of described in detail in this current study.
Author Response
Reviewer 4:
.1. This manuscript draws attention to an important topic in the effectiveness of technology integration and math education among teachers in special education.
We are very pleased that the reviewer emphasized the importance of the topic.
- Parts of this manuscript could be strengthened. For example, include more details on the rationale to develop a questionnaire for the current study. Are there any existing scales for these measures?
We added this information in the section addressing the development of the questionnaire.
- Please provide more detailed information regarding participants. Other than gender, years in teaching, and qualifications, any socio-demographic information was collected?
We did collect additional information on whether they followed some training on applications of technology in the classroom, and on applications of digital technologies in Math.
We included this information:
71.1% of the participants have taken between 1 to 3 courses on applications of digital technologies in education, 19.3% did not, while 9.6% reported to have participated in more than 3 courses. Specific training courses on applications of digital technologies in the Math class have been followed by 32.5% of the respondents. 67.5% did not take any course on this specific topic.
- Please provide more detailed information regarding the sample selection process and exclusion criteria, if any. When was data collection performed and completed?
We had a convenience sample. We did not use any exclusion criteria. We added, in the Procedure section information on the timeline on collecting the data.
“The questionnaire was administered online through Google forms, via email, between May and August 2022”.
- On p. 6, in Table 2, the reliability coefficient of .62 is quite low. Please add an explanation for this low reliability and item content.
One explanation for the lower value of the reliability coefficient is that the measure includes only three items. Another explanation is related to the content of the items, as each item samples beliefs on the limits of technology. For this reason, the correlation among the items may not be as high.
- On p. 8, line 319, please fill in the specific figures in the sentence.
We added the numbers. Thank you!
- Information in Tables 6 through 8 can be condensed into a short paragraph instead of described in detail in this current study.
While we thank you for this recommendation, we opted for keeping the table format which we fill facilitates the reading of this article.
Round 2
Reviewer 2 Report
I appreciate the efforts made by the authors to improve the paper according to the reviewers' suggestions.